# Dietary, Body Composition, and Blood Leptin Variations in Fit-Model Female Athletes During the Pre-Competition Period

**DOI:** 10.3390/nu17142299

**Published:** 2025-07-12

**Authors:** Ramutis Kairaitis, Petras Minderis, Inga Lukonaitienė, Gediminas Mamkus, Tomas Venckūnas, Sigitas Kamandulis

**Affiliations:** 1Department of Coaching Science, Lithuanian Sports University, 44221 Kaunas, Lithuania; 2Institute of Sports Science and Innovations, Lithuanian Sports University, 44221 Kaunas, Lithuania; petras.minderis@lsu.lt (P.M.); inga.lukonaitiene@lsu.lt (I.L.); gediminas.mamkus@lsu.lt (G.M.); tomas.venckunas@lsu.lt (T.V.); sigitas.kamandulis@lsu.lt (S.K.)

**Keywords:** calories, carbohydrate, fat, protein, leptin, menstrual cycle

## Abstract

**Background:** The Fit-Model in bodybuilding is a relatively new category designed for women seeking a balanced physique, avoiding excessive muscularity and extreme leanness. This study examined the dietary strategies, body composition changes, and plasma leptin fluctuations of Fit-Model athletes during a seven-week pre-competition phase. **Methods:** Twelve females (age: 27.6 ± 4.4 years, body mass: 60.0 ± 6.2 kg) preparing for a national championship were monitored for energy and macronutrient intakes, total, lean, and fat mass, plasma leptin levels, and menstrual cycle characteristics. The five highest-ranked athletes were selected to compete at the world championship, allowing for comparisons between national and international athletes. **Results:** Low carbohydrate intake was reported, and total energy intake decreased from 1700 to 1520 kcal/day approaching the contest day. Athletes experienced an average body mass loss of 4.2 kg, with no clear relationship between final weight or fat mass and competitive success. Plasma leptin levels were markedly low during all 7 weeks of preparation with a further decline before the contest, but did not correlate with either changes in body composition and weight or energy or macronutrient intakes. Menstrual cycle disturbances were prevalent, with only two athletes maintaining regular cycles by the end of the preparation. **Conclusions:** Fit-Model athletes undergo a considerable decline in body weight and fat mass during the final weeks before the contest, yet these changes do not appear to be decisive for performance outcomes. Persistently low leptin levels and menstrual irregularities call for strategies that balance physique optimization with endocrine health to support both the performance and well-being of athletes.

## 1. Introduction

The popularity of bodybuilding and fitness as both high-performance and recreational sports has increased significantly over the past two decades [1]. This growth reflects a broader cultural interest in strength training and its potential for improving health and physical appearance. In response, international bodybuilding federations have adapted by modifying existing divisions and introducing new categories to reflect evolving esthetic ideals and attract a broader participant base.

In early 2015, the International Fitness and Bodybuilding Federation (IFBB, https://ifbb.com/) introduced a new category, Fit-Model, to its competition program for females. This category, compared to other categories of fitness and especially bodybuilding, places less emphasis on muscularity and more on overall body symmetry, stage elegance, style, and physical attractiveness. The Fit-Model category is designed for women seeking a body shape that is neither excessively muscular nor overly lean [2]. Since its introduction, this category has rapidly become one of the most attended and popular in women’s bodybuilding competitions.

As with other bodybuilding categories, special interest in the Fit-Model category is given to the preparation period preceding competitions. The primary pre-competition goals for most bodybuilding categories are to reduce body fat while maintaining lean body mass (LBM). Common strategies to achieve these goals include reducing energy intake, increasing protein consumption, and incorporating aerobic training alongside strength training [3,4,5]. To preserve LBM, it is important to control the rate of body weight loss, aiming for a reduction of 0.5% to 1.0% of total body weight per week [6]. Additionally, a protein intake exceeding 2 g/kg/day is recommended to support muscle retention during caloric restriction [7,8].

Most existing studies on pre-competition tapering in esthetic physique sports focus on bodybuilding categories with greater muscularity than the Fit-Model category [9,10,11]. They also often overlook differences in category criteria across international federations. While pre-competition strategies have been studied in Bodybuilding, Physique, Classic Physique, and Figure [5,10,12,13,14,15,16,17], there is a notable lack of research on the Fit-Model division, despite its distinct esthetic judging criteria. This variability complicates the interpretation and comparison of research findings, particularly in women’s divisions, where the differences in categories are the largest. The IFBB World Federation ranks women’s categories by increasing muscularity: into Fit-Model, Bikini, Body Fitness, Wellness, and Physique, and the National Physique Committee (NPC) World Federation (https://npcnewsonline.com/) classifies women into Bikini, Figure, Physique, and Bodybuilding, following a similar muscularity progression. Comparing pre-competition preparation strategies across categories with vastly different muscularity requirements—such as Fit-Model versus Physique or Bikini versus Bodybuilding—is inappropriate and may lead to inaccurate conclusions.

Most existing studies on pre-competition preparation in esthetic physique sports focus on bodybuilding categories with greater muscularity than the Fit-Model category [9,10,11], often overlooking differences in category criteria across international federations. While pre-competition strategies have been studied in Bodybuilding, Physique, Classic Physique, and Figure [5,10,12,13,14,15,16,17], there is a notable lack of research on the Fit-Model division, despite its distinct esthetic judging criteria. This variability complicates the interpretation and comparison of findings, particularly in women’s divisions, where category differences are greatest. The IFBB ranks women’s categories by increasing muscularity as Fit-Model, Bikini, Body Fitness, Wellness, and Physique, while the National Physique Committee (NPC) World Federation (https://npcnewsonline.com/) uses Bikini, Figure, Physique, and Bodybuilding. Comparing preparation strategies across categories with vastly different muscularity requirements—such as Fit-Model versus Physique or Bikini versus Bodybuilding—is inappropriate and may lead to inaccurate conclusions.

Body shaping in Fit-Model athletes may be sensitive to subtle variations in hormonal and metabolic markers during contest preparation. One such marker is leptin, a hormone secreted predominantly by adipose tissue that plays a central role in regulating energy balance and body composition. Among the various hormones that fluctuate during caloric restriction—including ghrelin, testosterone, and estrogen—leptin is uniquely sensitive to reductions in adipose tissue mass and is therefore considered a key signal of energy availability and metabolic adaptation [18,19,20]. Unlike reproductive or anabolic hormones, which may be influenced by numerous unrelated factors (e.g., training intensity, supplementation, menstrual phase), leptin levels provide a more direct and reliable biomarker of adipose tissue loss and energy deficiency, especially in lean individuals preparing for competition. Moreover, changes in plasma leptin levels are tightly linked to menstrual cycle disruptions, which are common among female physique athletes due to energy restriction and low adipose tissue mass [21].

Although menstrual function is often overlooked in physique sport research, it provides a critical window into endocrine health and long-term well-being. Therefore, including menstrual status as an exploratory outcome in this study helps capture a more complete picture of physiological adaptations during competition preparation in the Fit-Model population.

The aim of this study was therefore to analyze the pre-competition preparation of Fit-Model athletes by examining their dietary practices (i.e., energy and macronutrient intake), changes in body weight and composition, plasma leptin levels, and menstrual cycle characteristics. We have also analyzed the correlation of changes in circulating leptin to those in energy and macronutrient intakes, and body mass components over the 7 weeks of the pre-competition period. By doing so, we aimed to contribute to a better understanding of the unique demands of this relatively new bodybuilding category and offer evidence-based insights for both competitive athletes and fitness enthusiasts pursuing similar esthetic goals. It was hypothesized that a combination of restricted energy intake and continued training during the final preparation phase for a major competition in already relatively lean athletes would lead to further reduction in body weight, largely due to adipose tissue loss, which would be accompanied by a profound drop in circulating leptin levels and menstrual cycle disruption. It was also anticipated that the reduction in plasma leptin levels in the overall group would be positively associated with a reduction in energy intake and body weight loss.

## 2. Materials and Methods

### 2.1. Participants

This study included 12 female athletes preparing intensively for a national Fit-Model championship (FM group; mean age: 27.6 ± 4.4 years, height: 165.8 ± 5.1 cm, body mass: 60.0 ± 6.2 kg). All followed the same pre-competition routine; therefore, dependent variables were analyzed across the entire group, with additional subgroup comparisons based on performance level. Athletes who advanced to the national finals and later competed at the world championship formed the International-level subgroup (FM-I; n = 5; mean age: 28.0 ± 6.7 years, training experience: 5.2 ± 0.4 years, competitive experience: 4.1 ± 0.5 years, height: 163.8 ± 4.8 cm, body mass: 58.8 ± 4.4 kg), while those who did not reach the finals were assigned to the National-level subgroup (FM-N; n = 7; mean age: 27.3 ± 2.1 years, training experience: 3.7 ± 1.8 years, competitive experience: 1.8 ± 1.4 years, height: 167.3 ± 5.1 cm, body mass: 60.9 ± 7.5 kg). Participants received detailed explanations of the study design, protocols, and potential risks. Every study participant confirmed that they were not using substances prohibited in their sport and were not taking contraceptive medication. Doping control was under implementation at both the Lithuanian championship and the world championship; that is, participants were aware that they could be tested for illicit substance usage.

### 2.2. Study Design

This study used an observational research design. The participants were monitored over seven weeks during the pre-competition period. At the end of each week, participants attended meetings with the full coaching staff to assess changes in body weight and appearance (body shape). Based on these assessments, expert coaches individually adjusted the training and dietary plans for the following week, prescribing energy and macronutrient intake for each day via specified food items and their distribution during the day. Detailed body mass analyses were conducted at the end of the first, fourth, and seventh weeks. On the same occasion, participants also completed questionnaires about their energy and macronutrient intakes over the preceding week, provided information on menstrual cycle regularity, gave blood samples, and received adjustments to their diet and exercise programs (see study design in Figure 1). The exercise selection was personalized to individual body-shaping goals, while all participants followed a similar routine consisting of 4–5 gym sessions per week, each lasting approximately 1.5 h with a primary focus on strength development. The study complied with the Declaration of Helsinki. Ethical approval was granted by the Lithuanian Sports University Biomedical Research Ethics Committee (No. BI-TRS (B)-2023-601, 13 April 2023). Before being enrolled in the study, all subjects read and signed an informed consent form. 

### 2.3. Data Collection

Participants attended the testing sessions in the early morning after fasting for 10–12 h. Minimal fluid intake (a glass of water) was permitted if they experienced thirst before measurements.

**Body Composition.** Total body mass was measured, and body fat mass and percentage, as well as lean body mass, were estimated using a bioimpedance-based body composition analyzer (Tanita, TBF-300, Arlington Heights, IL, USA). Measurements were made with participants in competitive attire and immediately after visiting the toilet.

**Nutrient Intake.** All participants were omnivorous. On three occasions (at the end of the first, fourth, and seventh weeks—Figure 1), they were completing, based on their notes, a questionnaire for exact consumption of prescribed daily foods, fluids and supplements over the preceding week, which was then reviewed and discussed for adherence with a team of expert coaches and one of the researchers. The questionnaire was based on the “Dietary Assessment of a Natural Bodybuilding Population” (http://bit.ly/3QqOrAW, accessed on 15 February 2023). Food consumption data were subsequently examined by the experienced researcher (PM), who determined energy and macronutrient intakes using the Cronometer software (https://cronometer.com). In the software, the reported food items were sourced from institutional databases such as the Nutrition Coordinating Center Food and Nutrient Database (NCCDB, https://nccdb.fmcsa.dot.gov/, accessed on 22 October 2023) and the United States Department of Agriculture National Nutrient Database for Standard Reference (USDA SR28, https://agdatacommons.nal.usda.gov/articles/dataset/USDA_National_Nutrient_Database_for_Standard_Reference_Legacy_Release/24661818, accessed on 22 October 2023), which provide the most comprehensive information on food energy and nutrient values.

**Menstrual Cycle.** All participants were asked to record the first day of menstruation throughout the entire 7-week period. During testing sessions, they were asked the following questions: (1) What is the normal length of your menstrual cycle? (2) What day of your cycle is today?

**Plasma leptin concentration analysis.** Venous blood was sampled from the median antecubital vein into 3 mL vacutainer tubes using EDTA with tri-potassium as an anticoagulant (K3 EDTA tube; Fisher Scientific, Waltham, MA, USA), inverted 8–10 times and kept at 2–8 °C until plasma was harvested by centrifugation at 1200× *g* for 15 min at 4 °C and stored in 0.5 mL aliquots at −80 °C until analysis. Plasma leptin concentration was analyzed by using a commercial ELISA kit (MD53001, IBL International GmbH, Hamburg, Germany) and a spectrophotometer plate reader (Spark 10M Tecan, Männedorf, Switzerland) at 450 nm wavelength.

### 2.4. Statistical Analysis

Data are expressed as mean ± standard deviation (SD). All data were normally distributed, as determined by visual inspection of histograms and measurements of skewness and kurtosis (normality assumed when values were between −2 and +2) [22].

A one-way repeated-measures ANOVA was conducted within the group (n = 12) to evaluate the effect of the independent variable pre-competition period—characterized by intensified training and dietary restrictions—as a time factor (T1, T2, T3) on the following dependent variables: protein, carbohydrate, fat and total energy intakes; body mass, lean body mass, body fat percent; and plasma leptin concentration. Bonferroni corrections were applied for pairwise comparisons. Additionally, changes between time points (T1 vs. T2, T2 vs. T3, T1 vs. T3) were analyzed separately in FM-N and FM-I subgroups to assess performance-related differences. Due to small sub-group samples, a nonparametric Friedman ANOVA test was used for these comparisons. Pearson’s correlation coefficient (r) determined the associations between changes in plasma leptin level and changes in other dependent variables within the group (n = 12) over T1–T3 and was interpreted as negligible (0–0.29), weak (0.30–0.49), moderate (0.50–0.69), strong (0.70–0.89), or very strong (0.90–1.00) [23]. Statistical analyses were performed using IBM SPSS Statistics 29.0 (IBM Corp., Armonk, NY, USA) with a significance level set at *p* < 0.05.

## 3. Results

### 3.1. Energy and Nutrient Intake

During the pre-competition period, participants reduced energy intake to an average of 1520 kcal (*p* < 0.05, Figure 2A). A significant reduction was observed only in FM-N participants (*p* < 0.05, Figure 2B), while both groups showed a similar pattern with no differences at any time point (T1, T2, T3).

The decrease in energy intake was primarily determined by a 35.6% reduction in carbohydrate consumption (1.75 to 1.13 g/kg/day, *p* < 0.05, Figure 3B). As competition approached, both groups tended to reduce fat intake (1.08 to 1.01 g/kg/day, *p* = 0.12, Figure 3E) and increase protein intake (2.70 to 3.06 g/kg/day, *p* = 0.07, Figure 3A). No significant differences were observed between FM-N and FM-I groups in carbohydrate, fat, or protein intake at any time point (T1–T3, *p* > 0.05, Figure 3B,D,E).

### 3.2. Body Composition and Leptin Levels

In both groups, body mass decreased during the pre-competition period (*p* < 0.001, Figure 4A,B), with an average loss of 4.2 kg. No significant differences were observed between groups, indicating a similar weight loss rate of ~1% per week. Lean body mass decreased slightly (1.2 kg, Figure 4C,D) but was not statistically significant, suggesting that weight loss was primarily due to a reduction in adipose tissue mass. Overall, adipose tissue mass decreased by 3.7% (*p* < 0.05, Figure 4E). By the end of the pre-competition period, the average body fat percentage was 20.0 ± 4.8% with no differences between FM-N and FM-I groups (Figure 4D).

Plasma leptin levels were several-fold lower than the normal range at the start of the pre-competition period (0.75 ng/ml vs. >3.5 ng/ml for women) and were higher in FM-N than in FM-I (*p* < 0.05, Figure 5A,B). Additionally, leptin levels further decreased over the pre-competition period (*p* = 0.018), mainly among FM-N participants. By the end of the period, no differences were observed between groups (*p* > 0.05).

### 3.3. Association Between Changes in Plasma Leptin Level with Changes in Food Intake and Body Composition

Changes in plasma leptin level were not significantly correlated with variation in any of the dependent variables (Table 1). However, plasma leptin level change showed a moderate positive correlation with body mass change, approaching statistical significance (*r* = 0.547, *p* = 0.065).

### 3.4. Menstrual Cycle

Among the 12 athletes, 11 had regular menstrual cycles before the preparation period (27.5 ± 1.7 days), while one had amenorrhea for 206 days. At the pre-competition phase, most were in the luteal (n = 7), while the others were in the follicular (n = 2) or menstrual (n = 2) phase. By the end of preparation, only 2 athletes retained normal menstrual cycles, while the others developed amenorrhea (n = 3), significantly shorter cycles (by 5 to 7 days; n = 3), or significantly longer cycles (by 3, 8, and 13 days).

## 4. Discussion

The findings of this study confirm significant changes in body composition and hormonal profiles among female athletes preparing for Fit-Model competitions. It is notable that initial (at T1 time point) body weight and fat mass were not associated with performance level, and reductions during the pre-competition period were consistent across individuals. This suggests that body composition alone does not determine performance outcomes in this bodybuilding category. Mainly, it was confirmed that other factors, such as genetic predisposition to body symmetry, as well as stage esthetic and elegance during performance, might play critical roles in athletic success [24].

The food intake analysis revealed a notable reduction in energy intake during the study, which, along with shifts in macronutrient distribution, were consistent with pre-competition preparation trends in other categories of bodybuilding. At the beginning of the pre-competition period, participants consumed an average of about 1700 kcal/day—a relatively low intake given their above-moderate physical activity and relatively large lean body mass. Participants further reduced their energy intake during the later stages of the preparation by approximately 13%. This value approaches the ones reported in athletes from other categories of bodybuilding during the pre-competition period, often referred to as the ‘cutting phase’ [15,25,26,27]. A systematic review by Helms et al. [6] found that competitive bodybuilders from other categories typically reduce energy intake by 15–25% in the final weeks. Compared to earlier established bodybuilding categories, slightly lower reduction in energy intake of our participants could have been due to specifics of the Fit-Model category, where other aspects of appearance than pronounced muscularity and its definition are relatively more important: females in Fit-Model category seek an “ideal” body shape that is neither overly lean nor excessively muscular [2]. Another reason for the relatively limited reduction in energy intake observed in our participants could be a lack of a broader perspective on their energy intake during the preparation period prior to the study and during the off-season.

In terms of macronutrient distribution, the current study observed a significantly reduced carbohydrate intake, which decreased by >35% in the total group. Research evidence that carbohydrate manipulation is a common practice in other bodybuilding categories to enhance fat loss while preserving lean muscle mass [5]. The findings suggest that strategic carbohydrate restriction can lead to improved body composition and performance metrics, which are particularly relevant for athletes in weight-sensitive and esthetic sports.

At the initial stages of preparation, large variations in carbohydrate intake (relative and body mass) in the FM-N group were observed. This variation could be related to lower experience in maintaining a strict diet, and may have subsequently led to larger variations in plasma leptin levels compared to the more experienced FM-I group. Dispersion of these two variables in the FM-N group subsided in later stages with coaching to further control diet. On the other hand, fat intake variation was more pronounced in the FM-I group throughout the investigation period. This suggests that the FM-I group had more experience and preferred nutritional strategies that guaranteed successful performance in previous preparations.

Additionally, both groups in the current study also reduced their fat intake by more than 13% as they approached the competitions. This coincides with the notion of the importance of reduced macronutrient intake in the final stages of esthetic competition preparation [28,29]. It is agreed that a well-structured dietary plan that includes reductions in both carbohydrates and fats can significantly enhance an athlete’s performance and esthetic goals.

Interestingly, the study found that both groups remained with about the same absolute protein intake, which led to the tendency to increase their relative protein intake from 2.70 g/kg/day to approximately 3.06 g/kg/day throughout the preparation period. This trend aligns with the recommendations made by Helms et al. [6], who advocate for high protein intake among bodybuilders of different categories to support muscle preservation during caloric deficits. Their findings suggest that maintaining a high protein intake is crucial for mitigating muscle loss while dieting, which is essential for athletes aiming to compete at a high level. Rukstela et al. [30] also emphasize that prioritizing protein over other macronutrients during the cutting phase is a smart practice, as it helps to spare muscle mass during energy deficits.

The study found that female Fit-Model competition participants experienced a significant reduction in body mass (4.2 kg) and estimated fat mass (3.0 kg) with a non-significant decrease in estimated lean body mass (1.2 kg) during preparation for competitions. The rate of ~1% per week of body mass reduction observed in the current study is consistent with findings from other studies that report similar rates of body mass loss in athletes [6]. Studies have shown that weight loss in both male and female physique athletes is associated with changes in plasma leptin levels, which may influence energy homeostasis and metabolic adaptations during the preparation for competitions [31,32]. As noted by Khan et al. [33], leptin plays a vital role in regulating body fat mass and energy expenditure, with its levels closely tied to the amount of adipose tissue present. While in the current study, leptin level variations did not correlate significantly with macronutrient intake and weight change, it should be noted that leptin levels were considerably below normal during the entire pre-competition period in our athletes. An additional decline was observed in the lower-performance group to reach the level of more experienced athletes, which had very low leptin concentrations from the beginning of the observation. Low leptin levels may indicate a pronounced and long-lasting energy deficit from previous dietary restrictions and contest preparation [34], especially in the higher-performance group in this study.

Menstrual irregularities were common in the pre-competition period, likely due to a combination of restricted energy intake and concomitant training. A significant reduction in energy intake (−1520 kcal), body mass (−4.2 kg), and adipose tissue mass (−3.0 kg) suggests low energy availability, a known cause of hypothalamic dysfunction and menstrual disturbances [35]. Leptin levels dropped markedly, potentially impairing gonadotropin-releasing hormone (GnRH) secretion [21]. Additionally, caloric restriction and intense training may have elevated cortisol, further suppressing GnRH and contributing to functional hypothalamic amenorrhea (FHA) [36]. These factors likely explain the amenorrhea (n = 3) and cycle alterations in others. Most of the athletes experienced menstrual cycle disruptions, which highlights the considerable impact of pre-competition preparation on endocrine and reproductive systems. The Fit-Model category of bodybuilding is a relatively new and thus rapidly developing sport; thus, it is reasonable to assume its rules will be changing, concomitantly exposing athletes to the new requirements, which will be asking for adjusted preparation needs both in terms of training and diet. This will probably reflect in the changes in physiological adaptations, including hormonal profile and menstrual perturbations. However, already at this stage of Fit-Model sports discipline evolvement, training and dietary practices currently undertaken by athletes in the proximity to their major stage performance, render circulating leptin levels very low and menstrual cycle disruptions prevalent, which in itself asks for a more careful tackling of the preparation.

*Limitations and Perspectives*. Small sample size of the athletes investigated and especially small sample sizes in different performance level subgroups preclude our findings from being of definitive translational value. Also, our subjects were supervised by a small number of closely related trainers/instructors, which additionally limits the generalizability of the findings. We also must admit that observation of athletes exclusively during their pre-competition phase and not during earlier training periods and/or during off-season, prevents us from disclosing the range of changes in body composition, macronutrient intakes, and plasma leptin levels these athletes were undergoing. All the above limitations call for more prolonged investigations with larger sample sizes in the future.

The study was limited to analyzing changes in macronutrient intakes, body weight, and composition, and menstrual cycle characteristics. Neither measurements of physiological parameters nor assessments of psychological state, such as vitality or mood, have been made, which leaves an open field for future studies. We have analyzed macronutrient and energy intakes, but did not elaborate on consumption of other important dietary components such as vitamins and minerals, the adequate intakes of which are also associated with well-being in the long term. In addition, we did not have a clear context of energy expenditures of the participants since neither their daily activity was properly quantified, nor was the basal or resting metabolic rate measured. Changes in both physical activity and resting metabolism would be particularly interesting to follow in the athletes of this popular sport. Finally, we had quantified body composition via bioelectrical impedance, which provides a relatively rough estimate of body fatness and additionally does not allow for segmental fat distribution, which is of practical importance for a vast majority of esthetic stage sports, including Fit-Model. Even if using bioelectrical impedance reflects changes in overall body composition relatively well [37], it does not allow for discrimination of subcutaneous vs. visceral adipose tissue content, which would be possible using more intricate methods for body composition analysis. All these limitations call for more in-depth investigations of this athletic population.

## 5. Conclusions

This study revealed that during the pre-competition period, female Fit-Model athletes undergo significant restriction of carbohydrate and total energy intakes, which are associated with reduced adipose tissue mass and total body weight. However, these changes in body composition do not appear to be the major determinants of stage performance success in their sports category. Detection of exceptionally low circulating leptin levels and prevalent menstrual disturbances highlight the need to better manage contest preparation to both optimize performance and avoid hormonal imbalance, to not compromising well-being.

## Figures and Tables

**Figure 1 nutrients-17-02299-f001:**
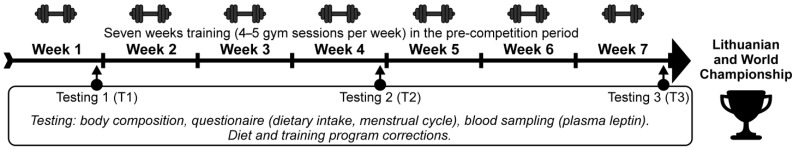
Design of the study.

**Figure 2 nutrients-17-02299-f002:**
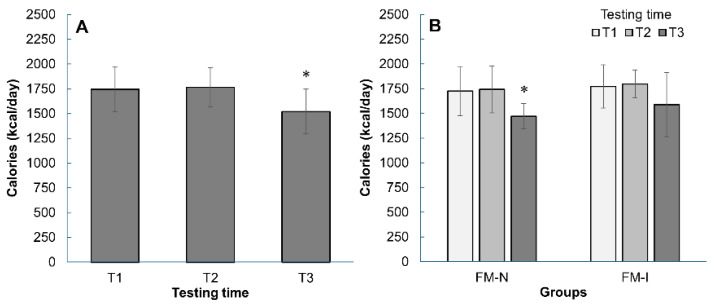
Energy intake changes during three measurements (T1, T2, and T3) in the FM-N and FM-I groups together (**A**) and separately (**B**). *—*p* < 0.05 compared with T2. Data are mean ± SD.

**Figure 3 nutrients-17-02299-f003:**
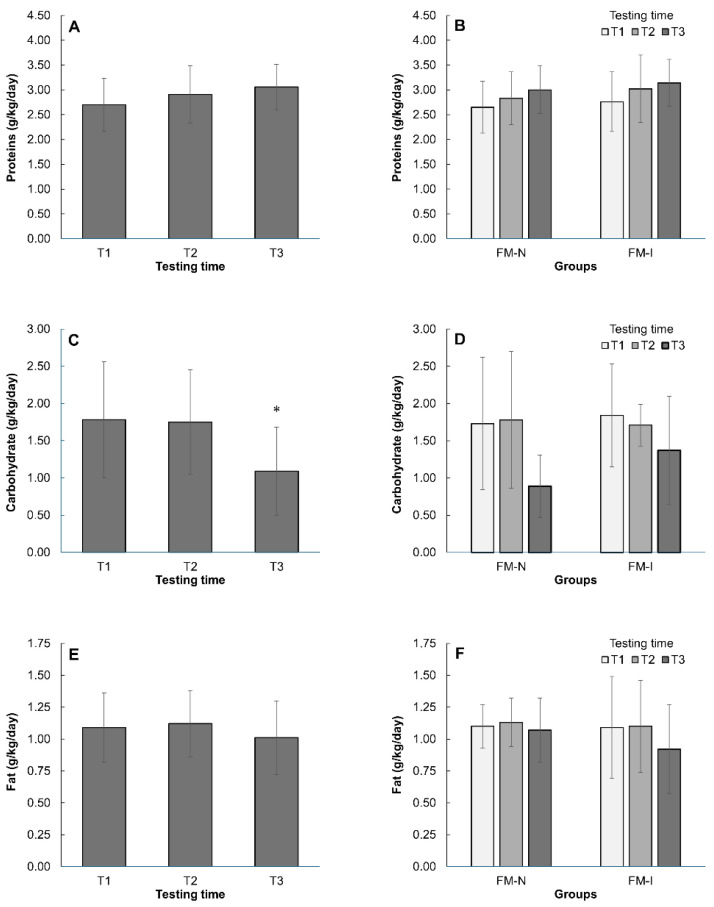
Protein, carbohydrate, and fat (g/kg/day) intake changes during three measurements (T1, T2, and T3) in the Fit-Model national (FM-N) and international (FM-I) groups together (**A**,**C**,**E**) and separately (**B**,**D**,**F**). *—*p* < 0.05 compared with T1 and T2. Data are mean ± SD.

**Figure 4 nutrients-17-02299-f004:**
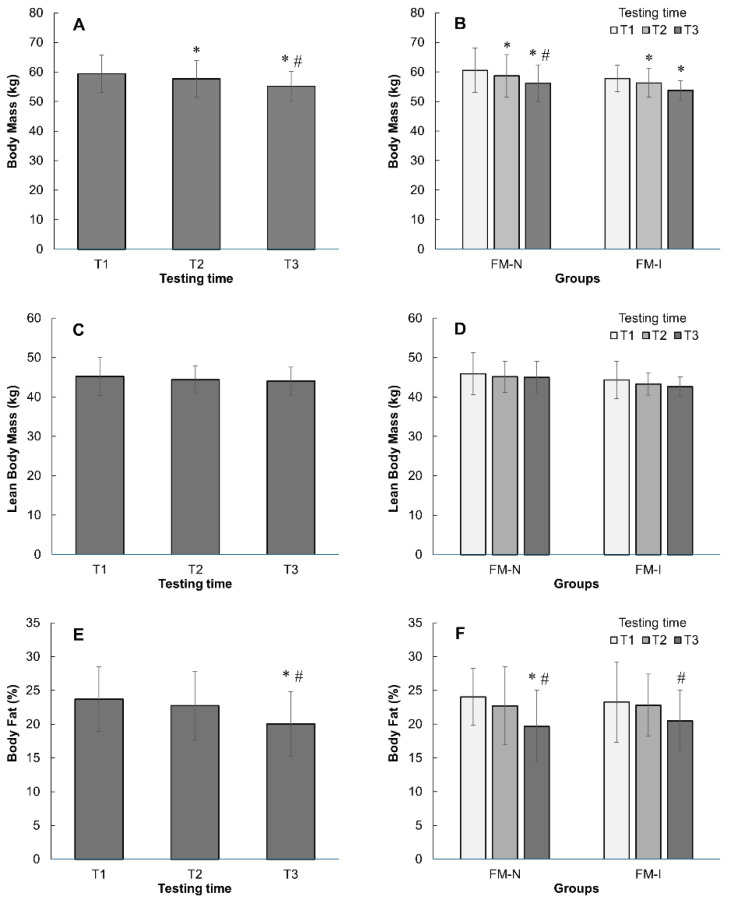
Body mass, lean body mass, and body fat % changes in Fit-Model national (FM-N) and international (FM-I) groups together (**A**,**C**,**E**) and separately (**B**,**D**,**F**). *—*p* < 0.05 in comparing with T1 of appropriate group; #—*p* < 0.05 compared with T2. Data are mean ± SD.

**Figure 5 nutrients-17-02299-f005:**
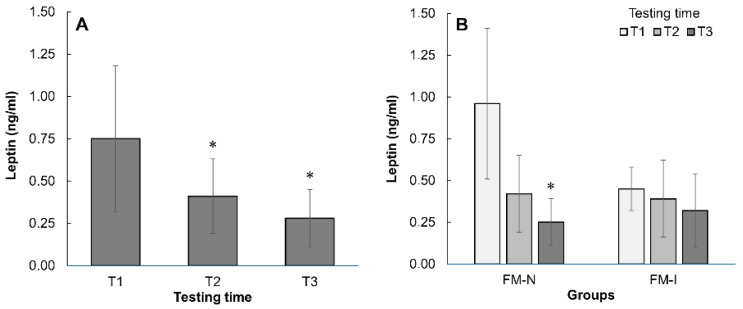
Leptin level changes in Fit-Model national (FM-N) and international (FM-I) groups together (**A**) and separately (**B**). *—*p* < 0.05 compared with T1. Data are mean ± SD.

**Table 1 nutrients-17-02299-t001:** Pearson correlation coefficients (*r*) between plasma leptin level change (T1–T3) and changes (T1–T3) in body weight, composition, and nutrient intakes.

	Body Weight	Lean Body Mass	Body Fat	
	(kg)	(kg)	(%)	(kg)
*r*	0.547	0.135	0.092	0.302
*p*	0.065	0.677	0.777	0.340
	**Calories**	**Protein**	**Carbohydrates**	**Fat**
	**(kcal/day)**	**(g/kg/day)**	**(g/kg/day)**	**(g/kg/day)**
*r*	0.252	−0.040	0.329	0.027
*p*	0.429	0.902	0.296	0.935

## Data Availability

Raw data are available upon reasonable request to the corresponding author.

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
