# Peer review of "Dietary, Body Composition, and Blood Leptin Variations in Fit-Model Female Athletes During the Pre-Competition Period"

_nutrients, 2025, doi:10.3390/nu17142299_

Round 1

Reviewer 1 Report

Comments and Suggestions for Authors

an interesting study, here are some comments, questions, and suggestions:

in the abstract - background section - should note why blood leptin variations or perhaps in the method section - since energy intake, macronutrient distribution, body mass, lean mass, fat mass seems a little bit known - blood leptin might be more medical

sampling? additionally in the background should also note why pre-competititon (though this was noted in the introduction which is good)

"ore-competitive tapering issues" is ore- a typing error

line 62 - for eample..... citation might be needed, similar to line 64 - in contrast...

might need to expand a bit more, why dietary practices, changes in body composition, and hormonal fluctuations are critical factors in the study

sampling? ethics approval for the study? consent form?

what now - conclusion could be expanded, what other practical implications can be provided?

are there any policy or competition rules implications? or any other issues that can be waranted or derived from the results

Reviewer 2 Report

Comments and Suggestions for Authors

Abstract: Please rewrite abstract after addressing all comments

Introduction: 

  • Please discuss leptin - lead the reader into understanding why this hormone must be looked at vs other hormones like ghrelin or testosterone, estrogen, etc... that can be tracked during an 8 week prep
  • Please correct spelling and grammar throughout the introduction
  • Please discuss why Fit-Model differs from these - where is the 'little evidence' you discuss in line 71? Please flush out why this division deserves attention and why the variables you're pointing out matter
  • The aim does not match the methods and results presented. You have secondary analyses, but you state you would like to focus on body comp, hormones (you've only done leptin), and dietary practices (you've only focused on nutrient intake and kcals). This must be clarified
  • Where is the hypothesis?
  • The entire introduction must flow better and discuss menstrual changes during prep as well

Methods

  • Restructure the entire methods to match the aims presented at the end of the introduction
  • How long have these 12 athletes been competing for? Are they all natural or have experience with PEDs? Please describe the sample with more detail
  • Where is the ethical statement sentence? I see it at the end of the paper but this highly concerning if not listed in methods
  • Table 1 is confusing -this should be figure and why do you mention 8 weeks and not 7 weeks in the methods? Is this an 8 week prep or 7 week? 
  • Where are baseline measures, excluding time point 1's weekend assessments? Where there any done, why/why not presented? Understanding these athletes baselines are important and will provide context for T1, T2 and T3
  • Data collection section - match the aims please
  • Body comp - body fat.. what? percent or mass? Please specify
  • Nutrient intake - you mentioned diet in the aims, yet no information on whether they were on keto, paleo, vegan, vegetarian, ETC.. please clarify what exactly you are measuring and discuss in the methods data collection section. If nutrients only, please make that clear throughout the entire paper
  • Where is the basal metabolic rates and the ranges per each macro for the sample of 12 athletes? How much did they receive or were aiming to receive (what is the range?), how were these determined and assigned? 
  • Please spell out all acronyms before referring to them 
  • If you're going to ask questions about the menstrual cycle - please discuss this importance in the introduction. This is not the same as 'hormonal fluctuations'. It seems these questions received less thought while planning the paper out

Statistical analysis

  • These statistical analyses do not match the intended aim. They seem more exploratory with an added correlational analysis; this must be corrected 
  • Please write out each dependent variable and independent variable, and the analysis used to show how objectives helped shape the analysis, and more importantly, how they align
  • leptin ... what units? concentrations of leptin?
  • Please rewrite this section so it clearly depicts what was analyzed and how it follows the aims

Results

  • Once again, the Section titles do not logistically follow the aim nor the methods. Please rewrite intro, methods, and results
  • What are the error bars on each figure, standard error or standard deviation? Please clarify, and please discuss why the variance is large for Figure 2D, Figure 2 F, Figure 4 B. None of these observations were discussed; however, they suggest some very important things 
  • Suggestions should be in the discussion section, the results are just that, results. Analysis is in the discussion

Discussion

  • Please rewrite the entire discussion to highlight the main findings first, followed by the aims of this paper. Currently as written does not present the same logic as the introduction
  • You mention basal metabolic rate but did not report it in methods... please report if you are going to discuss
  • Please fix grammatical errors and spelling throughout the discussion
  • Please be specific if citing Helms's study - you are providing more information for the Fit-Model division, how does this comparison (with bodybuilders) help, when this present division is a softer more feminine look? Please find other studies to compare with to make your point
  • 232-233 sentence: why is it slightly lower? Your graph shoes FM-N had higher variance in leptin and carbs than FM-I - please discuss this and prove the point you are trying to make 
  • 238: Once again, this is Fit Model division, please analyze for this division, present other research looking at this division, and not generalize it to other bodybuilding divisions 
  • 255-272 discusses leptin, but once again, there is no discussion of leptin in the introduction. please add this to the intro
  • 271-272: you need to report how many years these participants have been competing as this influences their baseline levels of everything (not just leptin)
  • Where is your limitations paragraph? You are missing detailed demographic information on the 12 women, detailed nutritional analysis and lean body mass/body fat %'s. Also, where are these women from? Did they have to travel? there are many factors that can affect performance than those presented here - please discuss those here so the reader understandings that this observational study is limited in its generalizations to other populations (aka, US observations in Fit Model)
  • Limitations: in the intro, you mention understanding the unique requirements of this category, yet say how similar this category is to others in the discussion - so are there limitations as to why you couldn't show how unique this division is, or are they even unique? It seems they are not based on your discussion. Please discuss how the sample and analysis supports this, or how it lacks to support this because of needing more information or longitudinal studies 
  • The last paragraph on menstrual irregularities is necessary, but it is not an aim in the introduction - once again, adjust the intro's aims and fix the methods/results

Once these sections are complete, please rewrite the abstract so the logic is consistent throughout

Comments on the Quality of English Language

This is a very important topic, especially for those interested in competing. However, significant information that is needed to fully communicate what was observed is missing. There are also too many grammatical and spelling errors throughout the paper. The paper could easily confuse someone unfamiliar to this sport, and undermine the importance of prep, especially hormonal changes. Please keep this in mind while rewriting this paper and addressing the comments I provided. They are there to help clarify the parameters for the reader so that they understand its context. 

Round 2

Reviewer 2 Report

Comments and Suggestions for Authors

Please see the attached pdf with comments dispersed throughout this second revision. 

Comments on the Quality of English Language

Attached a pdf where corrections are needed through comments. 
